# PointCNN: Convolution On $\mathcal{X}$-Transformed Points

**Yangyan Li**[†*]   **Rui Bu**[†]   **Mingchao Sun**[†]   **Wei Wu**[†]   **Xinhan Di**[‡]   **Baoquan Chen**[§]

[†]**Shandong University**         [‡]**Huawei Inc.**         [§]**Peking University**

## Abstract

We present a simple and general framework for feature learning from point clouds. The key to the success of CNNs is the convolution operator that is capable of leveraging spatially-local correlation in data represented densely in grids (e.g. images). However, point clouds are irregular and unordered, thus directly convolving kernels against features associated with the points will result in desertion of shape information and variance to point ordering. To address these problems, we propose to learn an $\mathcal{X}$-transformation from the input points to simultaneously promote two causes: the first is the weighting of the input features associated with the points, and the second is the permutation of the points into a latent and potentially canonical order. Element-wise product and sum operations of the typical convolution operator are subsequently applied on the $\mathcal{X}$-transformed features. The proposed method is a generalization of typical CNNs to feature learning from point clouds, thus we call it *PointCNN*. Experiments show that PointCNN achieves on par or better performance than state-of-the-art methods on multiple challenging benchmark datasets and tasks.

## 1   Introduction

Spatially-local correlation is a ubiquitous property of various types of data that is independent of the data representation. For data that is represented in regular domains, such as images, the convolution operator has been shown to be effective in exploiting that correlation as the key contributor to the success of CNNs on a variety of tasks [25]. However, for data represented in point cloud form, which is irregular and unordered, the convoralution operator is ill-suited for leveraging spatially-local correlations in the data.

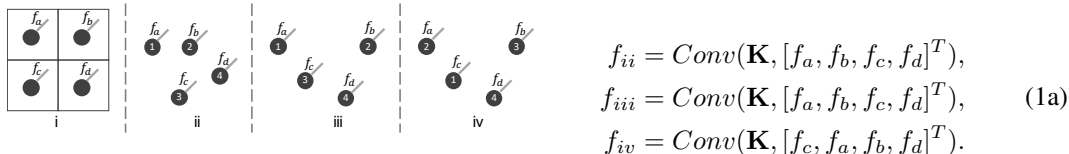

$$f_{ii} = Conv(\mathbf{K}, [f_a, f_b, f_c, f_d]^T),$$
$$f_{iii} = Conv(\mathbf{K}, [f_a, f_b, f_c, f_d]^T), \quad (1a)$$
$$f_{iv} = Conv(\mathbf{K}, [f_c, f_a, f_b, f_d]^T).$$

Figure 1: Convolution input from regular grids (i) and point clouds (ii-iv). In (i), each grid cell is associated with a feature. In (ii-iv), the points are sampled from local neighborhoods, in analogy to local patches in (i), and each point is associated with a feature, an order index, and coordinates.

$$f_{ii} = Conv(\mathbf{K}, \mathcal{X}_{ii} \times [f_a, f_b, f_c, f_d]^T),$$
$$f_{iii} = Conv(\mathbf{K}, \mathcal{X}_{iii} \times [f_a, f_b, f_c, f_d]^T), \quad (1b)$$
$$f_{iv} = Conv(\mathbf{K}, \mathcal{X}_{iv} \times [f_c, f_a, f_b, f_d]^T).$$

We illustrate the problems and challenges of applying convolutions on point clouds in Figure 1. Suppose the unordered set of the $C$-dimensional input features is the same $\mathbb{F} = \{f_a, f_b, f_c, f_d\}$

in all the cases (($i$) − ($iv$)), and we have one kernel $\mathbf{K} = [k_\alpha, k_\beta, k_\gamma, k_\delta]^T$ of shape $4 \times C$. In ($i$), by following the canonical order given by the regular grid structure, the features in the local $2 \times 2$ patch can be cast into $[f_a, f_b, f_c, f_d]^T$ of shape $4 \times C$, for convolving with $\mathbf{K}$, yielding $f_i = Conv(\mathbf{K}, [f_a, f_b, f_c, f_d]^T)$, where $Conv(\cdot, \cdot)$ is simply an element-wise product followed by a sum[2]. In ($ii$), ($iii$), and ($iv$), the points are sampled from local neighborhoods, and thus their ordering may be arbitrary. By following orders as illustrated in the figure, the input feature set $\mathbb{F}$ can be cast into $[f_a, f_b, f_c, f_d]^T$ in ($ii$) and ($iii$), and $[f_c, f_a, f_b, f_d]^T$ in (iv). Based on this, if the convolution operator is directly applied, the output features for the three cases could be computed as depicted in Eq. 1a. Note that $f_{ii} \equiv f_{iii}$ holds for all cases, while $f_{iii} \neq f_{iv}$ holds for most cases. This example illustrates that a direct convolution results in deserting shape information (i.e., $f_{ii} \equiv f_{iii}$), while retaining variance to the ordering (i.e., $f_{iii} \neq f_{iv}$).

In this paper, we propose to learn a $K \times K$ $\mathcal{X}$-transformation for the coordinates of $K$ input points $(p_1, p_2, ..., p_K)$, with a multilayer perceptron [39], i.e., $\mathcal{X} = MLP(p_1, p_2, ..., p_K)$. Our aim is to use it to simultaneously weight and permute the input features, and subsequently apply a typical convolution on the transformed features. We refer to this process as $\mathcal{X}$-Conv, and it is the basic building block for our PointCNN. The $\mathcal{X}$-Conv for ($ii$), ($iii$), and ($iv$) in Figure 1 can be formulated as in Eq. 1b, where the $\mathcal{X}$s are $4 \times 4$ matrices, as $K = 4$ in this figure. Note that since $\mathcal{X}_{ii}$ and $\mathcal{X}_{iii}$ are learned from points of different shapes, they can differ so as to weight the input features accordingly, and achieve $f_{ii} \neq f_{iii}$. For $\mathcal{X}_{iii}$ and $\mathcal{X}_{iv}$, if they are learned to satisfy $\mathcal{X}_{iii} = \mathcal{X}_{iv} \times \Pi$, where $\Pi$ is the permutation matrix for permuting $(c, a, b, d)$ into $(a, b, c, d)$, then $f_{iii} \equiv f_{iv}$ can be achieved.

From the analysis of the example in Figure 1, it is clear that, with ideal $\mathcal{X}$-transformations, $\mathcal{X}$-Conv is capable of taking the point shapes into consideration, while being invariant to ordering. In practice, we find that the learned $\mathcal{X}$-transformations are far from ideal, especially in terms of the permutation equivalence aspect. Nevertheless, PointCNN built with $\mathcal{X}$-Conv is still significantly better than a direct application of typical convolutions on point clouds, and on par or better than state-of-the-art neural networks designed for point cloud input data, such as PointNet++ [35].

Section 3 contains the details of $\mathcal{X}$-Conv, as well as PointCNN architectures. We show our results on multiple challenging benchmark datasets and tasks in Section 4, together with ablation experiments and visualizations for a better understanding of PointCNN.

## 2 Related Work

**Feature Learning from Regular Domains.** CNNs have been very successful for leveraging spatially-local correlation in images — pixels in 2D regular grids [26]. There has been work in extending CNNs to higher dimensional regular domains, such as 3D voxels [52]. However, as both the input and convolution kernels are of higher dimensions, the amount of both computation and memory inflates dramatically. Octree [37, 47], Kd-Tree [22] and Hash [41] based approaches have been proposed to save computation by skipping convolution in empty space. The activations are kept sparse in [13] to retain sparsity in convolved layers. [17] and [4] partition point cloud into grids and represent each grid with grid mean points and Fisher vectors respectively for convolving with 3D kernels. In these approaches, the kernels themselves are still dense and of high dimension. Sparse kernels are proposed in [28], but this approach cannot be applied recursively for learning hierarchical features. Compared with these methods, PointCNN is sparse in both input representation and convolution kernels.

**Feature Learning from Irregular Domains.** Stimulated by the rapid advances and demands in 3D sensing, there has been quite a few recent developments in feature learning from 3D point clouds. PointNet [33] and Deep Sets [58] proposed to achieve input order invariance by the use of a symmetric function over inputs. PointNet++ [35] and SO-Net [27] apply PointNet hierarchically for better capturing of local structures. Kernel correlation and graph pooling are proposed for improving PointNet-like methods in [42]. RNN is used in [18] for processing features aggregated by pooling from ordered point cloud slices. [50] proposed to leverage neighborhood structures in both point and feature spaces. While these symmetric pooling based approaches, as well as those in [10, 58, 36], have guarantee in achieving order invariance, they come with a price of throwing away information.

[43, 3, 44] propose to first "interpolate" or "project" features into *predefined* regular domains, where typical CNNs can be applied. In contrast, the regular domain is latent in our method. CNN kernels are represented as parametric functions of neighborhood point positions to generalize CNNs for point clouds in [48, 14, 53]. The kernels associated with each point are parametrized individually in these methods, while the $\mathcal{X}$-transformations in our method are learned from each neighborhood, thus could potentially by more adaptive to local structures.

Besides as point clouds, sparse data in irregular domains can be represented as graphs, or meshes, and a few works have been proposed for feature learning from such representations [31, 55, 30]. We refer the interested reader to [5] for a comprehensive survey of work along these directions. Spectral graph convolution on a local graph is used for processing point clouds in [46].

**Invariance vs. Equivariance.** A line of pioneering work aiming at achieving equivariance has been proposed to address the information loss problem of pooling in achieving invariance [16, 40]. The $\mathcal{X}$-transformations in our formulation, ideally, are capable of realizing equivariance, and are demonstrated to be effective in practice. We also found similarity between PointCNN and Spatial Transformer Networks [20], in the sense that both of them provided a mechanism to "transform" input into latent canonical forms for being further processed, with no explicit loss or constraint in enforcing the canonicalization. In practice, it turns out that the networks find their ways to leverage the mechanism for learning better. In PointCNN, the $\mathcal{X}$-transformation is supposed to serve for both weighting and permutation, thus is modelled as a general matrix. This is different than that in [8], where a permutation matrix is the desired output, and is approximated by a doubly stochastic matrix.

## 3 PointCNN

The hierarchical application of convolutions is essential for learning hierarchical representations via CNNs. PointCNN shares the same design and generalizes it to point clouds. First, we introduce hierarchical convolutions in PointCNN, in analogy to that of image CNNs, then, we explain the core $\mathcal{X}$-Conv operator in detail, and finally, present PointCNN architectures geared toward various tasks.

### 3.1 Hierarchical Convolution

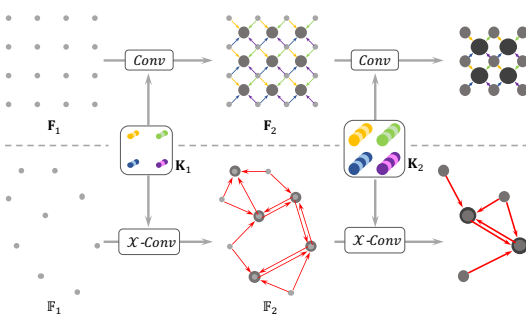

Figure 2: Hierarchical convolution on regular grids (upper) and point clouds (lower). In regular grids, convolutions are recursively applied on local grid patches, which often reduces the grid resolution ($4 \times 4 \rightarrow 3 \times 3 \rightarrow 2 \times 2$), while increasing the channel number (visualized by dot thickness). Similarly, in point clouds, $\mathcal{X}$-Conv is recursively applied to "project", or "aggregate", information from neighborhoods into fewer representative points ($9 \rightarrow 5 \rightarrow 2$), but each with richer information.

Before we introduce the hierarchical convolution in PointCNN, we briefly go through its well known version for regular grids, as illustrated in Figure 2 upper. The input to grid-based CNNs is a feature map $\mathbf{F}_1$ of shape $R_1 \times R_1 \times C_1$, where $R_1$ is the spatial resolution, and $C_1$ is the feature channel depth. The convolution of kernels $\mathbf{K}$ of shape $K \times K \times C_1 \times C_2$ against local patches of shape $K \times K \times C_1$ from $\mathbf{F}_1$, yields another feature map $\mathbf{F}_2$ of shape $R_2 \times R_2 \times C_2$. Note that in Figure 2 upper, $R_1 = 4$, $K = 2$, and $R_2 = 3$. Compared with $\mathbf{F}_1$, $\mathbf{F}_2$ is often of lower resolution ($R_2 < R_1$) and of deeper channels ($C_2 > C_1$), and encodes higher level information. This process is recursively applied, producing feature maps with decreasing spatial resolution ($4 \times 4 \rightarrow 3 \times 3 \rightarrow 2 \times 2$ in Figure 2 upper), but deeper channels (visualized by increasingly thicker dots in Figure 2 upper).

The input to PointCNN is $\mathbb{F}_1 = \{(p_{1,i}, f_{1,i}) : i = 1, 2, ..., N_1\}$, i.e., a set of points $\{p_{1,i} : p_{1,i} \in \mathbb{R}^{Dim}\}$, each associated with a feature $\{f_{1,i} : f_{1,i} \in \mathbb{R}^{C_1}\}$. Following the hierarchical construction of grid-based CNNs, we would like to apply $\mathcal{X}$-Conv on $\mathbb{F}_1$ to obtain a higher level representation $\mathbb{F}_2 = \{(p_{2,i}, f_{2,i}) : f_{2,i} \in \mathbb{R}^{C_2}, i = 1, 2, ..., N_2\}$, where $\{p_{2,i}\}$ is a set of representative points of

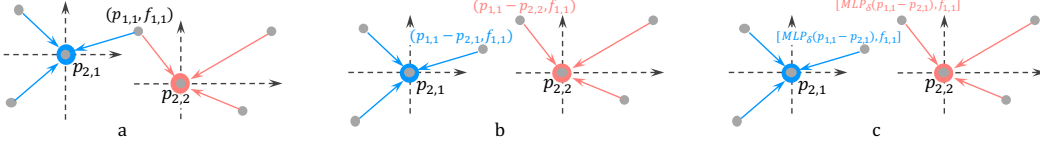

Figure 3: The process for converting point coordinates to features. Neighboring points are transformed to the local coordinate systems of the representative points (a and b). The local coordinates of each point are then individually lifted and combined with the associated features (c).

$\{p_{1,i}\}$ and $\mathbb{F}_2$ is of a smaller spatial resolution and deeper feature channels than $\mathbb{F}_1$, i.e., $N_2 < N_1$, and $C_2 > C_1$. When the $\mathcal{X}$-Conv process of turning $\mathbb{F}_1$ into $\mathbb{F}_2$ is recursively applied, the input points with features are "projected", or "aggregated", into fewer points ($9 \rightarrow 5 \rightarrow 2$ in Figure 2 lower), but each with increasingly richer features (visualized by increasingly thicker dots in Figure 2 lower).

The representative points $\{p_{2,i}\}$ should be the points that are beneficial for the information "projection" or "aggregation". In our implementation, they are generated by random down-sampling of $\{p_{1,i}\}$ in classification tasks, and farthest point sampling in segmentation tasks, since segmentation tasks are more demanding on a uniform point distribution. We suspect some more advanced point selections which have shown promising performance in geometry processing, such as Deep Points [51], could fit in here as well. We leave the exploration of better representative point generation methods for future work.

## 3.2 $\mathcal{X}$-Conv Operator

$\mathcal{X}$-Conv is the core operator for turning $\mathbb{F}_1$ into $\mathbb{F}_2$. In this section, we first introduce the input, output and procedure of the operator, and then explain the rationale behind the procedure.

---

**ALGORITHM 1: $\mathcal{X}$-Conv Operator**

---

**Input** $\;:\mathbf{K}, p, \mathbf{P}, \mathbf{F}$

**Output :** $\mathbf{F}_p$        ▷ Features "projected", or "aggregated", into representative point $p$

1: $\mathbf{P}' \leftarrow \mathbf{P} - p$        ▷ Move $\mathbf{P}$ to local coordinate system of $p$

2: $\mathbf{F}_\delta \leftarrow MLP_\delta(\mathbf{P}')$        ▷ **Individually** lift each point into $C_\delta$ dimensional space

3: $\mathbf{F}_* \leftarrow [\mathbf{F}_\delta, \mathbf{F}]$        ▷ Concatenate $\mathbf{F}_\delta$ and $\mathbf{F}$, $\mathbf{F}_*$ is a $K \times (C_\delta + C_1)$ matrix

4: $\mathcal{X} \leftarrow MLP(\mathbf{P}')$        ▷ Learn the $K \times K$ $\mathcal{X}$-transformation matrix

5: $\mathbf{F}_\mathcal{X} \leftarrow \mathcal{X} \times \mathbf{F}_*$        ▷ Weight and permute $\mathbf{F}_*$ with the learnt $\mathcal{X}$

6: $\mathbf{F}_p \leftarrow \text{Conv}(\mathbf{K}, \mathbf{F}_\mathcal{X})$        ▷ Finally, typical convolution between $\mathbf{K}$ and $\mathbf{F}_\mathcal{X}$

---

To leverage spatially-local correlation, similar to convolution in grid-based CNNs, $\mathcal{X}$-Conv operates in local regions. Since the output features are supposed to be associated with the representative points $\{p_{2,i}\}$, $\mathcal{X}$-Conv takes their neighborhood points in $\{p_{1,i}\}$, as well as the associated features, as input to convolve with. For simplicity, we denote a representative point in $\{p_{2,i}\}$ as $p$, the features with $p$ as $f$ and its $K$ neighbors in $\{p_{1,i}\}$ as $\mathbb{N}$, thus the $\mathcal{X}$-Conv input for this specific $p$ is $\mathbb{S} = \{(p_i, f_i) : p_i \in \mathbb{N}\}$. Note that $\mathbb{S}$ is an unordered set. Without loss of generality, $\mathbb{S}$ can be cast into a $K \times Dim$ matrix $\mathbf{P} = (p_1, p_2, ..., p_K)^T$, and a $K \times C_1$ matrix $\mathbf{F} = (f_1, f_2, ..., f_K)^T$, and $\mathbf{K}$ denotes the trainable convolution kernels. With these inputs, we would like to compute the features $\mathbf{F}_p$, which are the "projection", or "aggregation", of the input features into the representative point $p$. We detail the $\mathcal{X}$-Conv operator in Algorithm 1, and summarize it concisely as:

$$\mathbf{F}_p = \mathcal{X}-Conv(\mathbf{K}, p, \mathbf{P}, \mathbf{F}) = Conv(\mathbf{K}, MLP(\mathbf{P} - p) \times [MLP_\delta(\mathbf{P} - p), \mathbf{F}]), \quad (2)$$

where $MLP_\delta(\cdot)$ is a multilayer perceptron applied individually on each point, as in PointNet [33]. Note that all the operations involved in building $\mathcal{X}$-Conv, i.e., $Conv(\cdot, \cdot)$, $MLP(\cdot)$, matrix multiplication $(\cdot) \times (\cdot)$, and $MLP_\delta(\cdot)$, are differentiable. Accordingly. $\mathcal{X}$-Conv is differentiable, and can be plugged into a neural network for training by back propagation.

Lines 4-6 in Algorithm 1 are the core $\mathcal{X}$-transformation as described in Eq. 1b in Section 1. Here, we explain the rationale behind lines 1-3 of Algorithm 1 in detail. $\mathcal{X}$-Conv is designed to work on local point regions, and the output should not be dependent on the absolute position of $p$ and

its neighboring points, but on their relative positions. To that end, we position local coordinate systems at the representative points (line 1 of Algorithm 1, Figure 3b). It is the local coordinates of neighboring points, together with their associated features, that define the output features. However, the local coordinates are of a different dimensionality and representation than the associated features. To address this issue, we first lift the coordinates into a higher dimensional and more abstract representation (line 2 of Algorithm 1), and then combine it with the associated features (line 3 of Algorithm 1) for further processing (Figure 3c).

Lifting coordinates into features is done through a point-wise $MLP_\delta(\cdot)$, as in PointNet-based methods. Differently, however, the lifted features are not processed by a symmetric function. Instead, along with the associated features, they are weighted and permuted by the $\mathcal{X}$-transformation that is jointly learned across all neighborhoods. The resulting $\mathcal{X}$ **is** dependent on the order of the points, and this **is** desired, as $\mathcal{X}$ is supposed to permute $\mathbf{F}_*$ according to the input points, and therefore has to be aware of the specific input order. For an input point cloud without any additional features, i.e., $\mathbf{F}$ is empty, the first $\mathcal{X}$-Conv layer uses only $\mathbf{F}_\delta$. PointCNN can thus handle point clouds with or without additional features in a robust uniform fashion.

For more details about the $\mathcal{X}$-Conv operator, including the actual definition of $MLP_\delta(\cdot)$, $MLP(\cdot)$ and $\mathrm{Conv}(\cdot, \cdot)$, please refer to Supplementary Material Section 1.

### 3.3 PointCNN Architectures

From Figure 2, we can see that the Conv layers in grid-based CNNs and $\mathcal{X}$-Conv layers in PointCNN only differ in two aspects: the way the local regions are extracted ($K \times K$ patches vs. $K$ neighboring points around representative points) and the way the information from local regions is learned (Conv vs. $\mathcal{X}$-Conv). Otherwise, the process of assembling a deep network with $\mathcal{X}$-Conv layers highly resembles that of grid-based CNNs.

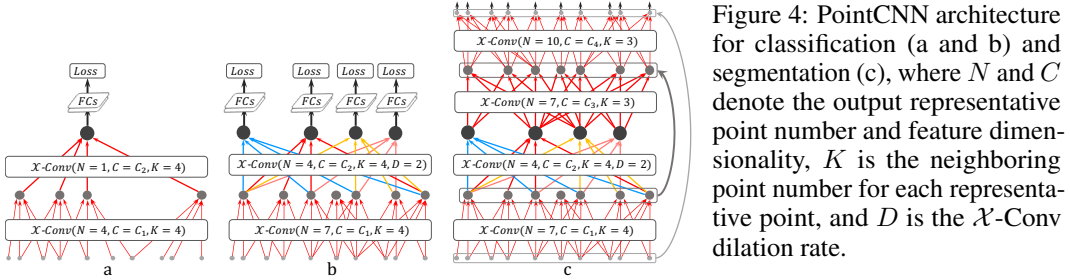

Figure 4: PointCNN architecture for classification (a and b) and segmentation (c), where $N$ and $C$ denote the output representative point number and feature dimensionality, $K$ is the neighboring point number for each representative point, and $D$ is the $\mathcal{X}$-Conv dilation rate.

Figure 4a depicts a simple PointCNN with two $\mathcal{X}$-Conv layers that gradually transform the input points (with or without features) into fewer representation points, but each with richer features. After the second $\mathcal{X}$-Conv layer, there is only one representative point left, and it aggregates information from all the points from the previous layer. In PointCNN, we can roughly define the receptive field of each representative point as the ratio $K/N$, where $K$ is the neighboring point number, and $N$ is the point number in the previous layer. With this definition, the final point "sees" all the points from the previous layer, thus has a receptive field of $1.0$ — it has a global view of the entire shape, and its features are informative for semantic understanding of the shape. We can add fully connected layers on top of the last $\mathcal{X}$-Conv layer output, followed by a loss, for training the network.

Note that the number of training samples for the top $\mathcal{X}$-Conv layers drops rapidly (Figure 4a), making it inefficient to train them thoroughly. To address this problem, we propose PointCNN with denser connections (Figure 4b), where more representative points are kept in the $\mathcal{X}$-Conv layers. However, we aim to maintain the depth of the network, while keeping the receptive field growth rate, such that the deeper representative points "see" increasingly larger portions of the entire shape. We achieve this goal by employing the dilated convolution idea from grid-based CNNs in PointCNN. Instead of always taking the $K$ neighboring points as input, we uniformly sample $K$ input points from $K \times D$ neighboring points, where $D$ is the dilation rate. In this case, the receptive field increases from $K/N$ to $(K \times D)/N$, without increasing actual neighboring point count or kernel size.

In the second $\mathcal{X}$-Conv layer of PointCNN in Figure 4b, dilation rate $D = 2$ is used, thus all the four remaining representative points "see" the entire shape, and all of them are suitable for making

| | ModelNet40 | | | | ScanNet | |
|---|---|---|---|---|---|---|
| | Pre-aligned | | Unaligned | | | |
| | mA | OA | mA | OA | mA | OA |
| Flex-Convolution [14] | - | 90.2 | - | - | - | - |
| KCNet [42] | - | 91 | - | - | - | - |
| Kd-Net [22] | 88.5 | 90.6 (91.8 w/ P32768) | - | - | - | - |
| SO-Net [27] | - | 90.7 (93.4 w/ PN5000) | - | - | - | - |
| 3DmFV-Net [4] | - | 91.4 (91.6 w/ P2048) | - | - | - | - |
| PCNN [3] | - | 92.3 | - | - | - | - |
| PointNet [33] | - | - | 86.2 | 89.2 | - | - |
| PointNet++ [35] | - | - | - | 90.7 (91.9 w/ PN5000) | - | 76.1 |
| SpecGCN [46] | - | - | - | 91.5 (92.1 w/ PN2048) | - | - |
| SpiderCNN [53] | - | - | - | - (92.4 w/ PN1024) | - | - |
| DGCNN [50] | - | - | **90.2** | 92.2 | - | - |
| PointCNN | **88.8** | **92.5** | 88.1 | 92.2 | 55.7 | 79.7 |

Table 1: Comparisons of mean per-class accuracy (mA) and overall accuracy (OA) (%) on ModelNet40 [52] and ScanNet [9]. The reported performances are based on $1024$ input points, unless otherwise noted by P# (# input points) or PN# (# input points with normals).

predictions. Note that, in this way, we can train the top $\mathcal{X}$-Conv layers more thoroughly, as much more connections are involved in the network, compared to PointCNN in Figure 4a. In test time, the output from the multiple representative points is averaged right before the $softmax$ to stabilize the prediction. This design is similar to that of Network in Network [29]. The denser version of PointCNN (Figure 4b) is the one we used for classification tasks.

For segmentation tasks, high resolution point-wise output is required, and this can be realized by building PointCNN following Conv-DeConv [32] architecture, where the DeConv part is responsible for propagating global information into high resolution predictions (see Figure 4c). Note that both the "Conv" and "DecConv" in the PointCNN segmentation network are the same $\mathcal{X}$-Conv operator. The only differences between the "Conv" and "DeConv" layers is that the latter has more points but less feature channels in its output vs. its input, and its higher resolution points are forwarded from earlier "Conv" layers, following the design of U-Net [38].

Dropout is applied before the last fully connected layer to reduce over-fitting. We also employed the "subvolume supervision" idea from [34], to further address the over-fitting problem. In the last $\mathcal{X}$-Conv layers, the receptive field is set to be less than 1, such that only partial information is "seen" by the representative points. The network is pushed to learn harder from the partial information during training, and performs better at test time. In this case, the global coordinates of the representative points matter, thus they are lifted into feature space $\mathbb{R}^{C_g}$ with $MLP_g(\cdot)$ (detailed in Supp. Material Section 1) and concatenated into $\mathcal{X}$-Conv for further processing by follow-up layers.

**Data augmentation.** To train the parameters in $\mathcal{X}$-Conv, it is evidently not beneficial to keep using the same set of neighboring points, in the same order, for a specific representative point. To improve generalization, we propose to randomly sample and shuffle the input points, such that both the neighboring point sets and order may differ from batch to batch. To train a model that takes $N$ points as input, $\mathcal{N}(N, (N/8)^2)$ points are used for training, where $\mathcal{N}$ denotes a Gaussian distribution. We found that this strategy is crucial for successful training of PointCNN.

## 4 Experiments

We conducted an extensive evaluation of PointCNN for shape classification on six datasets (Model-Net40 [52], ScanNet [9], TU-Berlin [11], Quick Draw [15], MNIST, CIFAR10), and segmentation task on three datasets (ShapeNet Parts [54], S3DIS [2], and ScanNet [9]). The details of the datasets and how we convert and feed data into PointCNN, are described in Supp. Material Section 2, and the PointCNN architectures for the tasks on these datasets can be found in Supp. Material Section 3.

### 4.1 Classification and Segmentation Results

We summarize our 3D point cloud classification results on ModelNet40 and ScanNet in Table 1, and compare to several neural network methods designed for point clouds. Note that a large portion of the 3D models from ModelNet40 are pre-aligned to the common up direction and horizontal facing direction. If a random horizontal rotation is not applied on either the training or testing sets, then the relatively consistent horizontal facing direction is leveraged, and the metrics based on this setting is not directly comparable to those with the random horizontal rotation. For this reason, we ran PointCNN and reported its performance in both settings. Note that PointCNN achieved top performance on both ModelNet40 and ScanNet.

We evaluate PointCNN on the segmentation of ShapeNet Parts, S3DIS, and ScanNet datasets, and summarize the results in Table 2. More detailed segmentation result comparisons can be found in Supplementary Material Section 4. We note that PointCNN outperforms all the compared methods, including SSCN [12], SP-Graph [24] and SGPN [49], which are specialized segmentation networks with state-of-the-art performance. Note that the part averaged IoU metric for ShapeNet Parts is the one used in [56]. Compared with mean IoU, the part averaged IoU puts more emphasis on the correct prediction of small parts.

Sketches are 1D curves in 2D space, thus can be more effectively represented with point clouds, rather than with 2D images. We evaluate PointCNN on TU-Berlin and Quick Draw sketches, and present results in Table 3, where we compare its performance with the competitive PointNet++, as well as image CNN based methods. PointCNN outperforms PointNet++ on both datasets, with a more prominent advantage on Quick Draw (25M data samples), which

| | ShapeNet Parts | | S3DIS | ScanNet |
|---|---|---|---|---|
| | pIoU | mpIoU | mIoU | OA |
| SyncSpecCNN [55] | 84.74 | 82.0 | - | - |
| Pd-Network [22] | 85.49 | 82.7 | - | - |
| SSCN [12] | 85.98 | 83.3 | - | - |
| SPLATNet [43] | 85.4 | 83.7 | - | - |
| SpiderCNN [53] | 85.3 | 81.7 | - | - |
| SO-Net [27] | 84.9 | 81.0 | - | - |
| PCNN [3] | 85.1 | 81.8 | - | - |
| KCNet [42] | 83.7 | 82.2 | - | - |
| SpecGCN [46] | 85.4 | - | - | - |
| Kd-Net [22] | 82.3 | 77.4 | - | - |
| 3DmFV-Net [4] | 84.3 | 81.0 | - | - |
| RSNet [18] | 84.9 | 81.4 | 56.47 | - |
| DGCNN [50] | 85.1 | 82.3 | 56.1 | - |
| PointNet [33] | 83.7 | 80.4 | 47.6 | 73.9 |
| PointNet++ [35] | 85.1 | 81.9 | - | 84.5 |
| SGPN [49] | 85.8 | 82.8 | 50.37 | - |
| SPGraph [24] | - | - | 62.1 | - |
| TCDP [44] | - | - | - | 80.9 |
| PointCNN | **86.14** | **84.6** | **65.39** | **85.1** |

Table 2: Segmentation comparisons on ShapeNet Parts in part-averaged IoU (pIoU, %) and mean per-class pIoU (mpIoU, %), S3DIS in mean per-class IoU (mIoU, %) and ScanNet in per voxel overall accuracy (OA, %).

is significantly larger than TU-Berlin (0.02M data samples). On the TU-Berlin dataset, while the performance of PointCNN is slightly better than the generic image CNN AlexNet [23], there is still a gap with the specialized Sketch-a-Net [57]. It is interesting to study whether architectural elements from Sketch-a-Net can be adopted and integrated into PointCNN to improve its performance on the sketch datasets.

Since $\mathcal{X}$-Conv is a generalization of Conv, ideally, PointCNN should perform on par with CNNs, if the underlying data is the same, but only represented differently. To verify this, we evaluate PointCNN on the point cloud representation of MNIST and CIFAR10, and show results in Table 4. For MNIST data, PointCNN achieved comparable performance with other methods, indicating its effective learning of the digits' shape information. For CIFAR10 data, where there is mostly no "shape" information, PointCNN has to learn mostly from the spatially-local correlation in the RGB features, and it performed reasonably well on this task, though there is a large gap between PointCNN and the mainstream image CNNs. From this experiment, we can conclude that CNNs are still the better choice for general images.

| Method | TU-Berlin | Quick Draw |
|---|---|---|
| Sketch-a-Net [57] | **77.95** | - |
| AlexNet [23] | 68.60 | - |
| PointNet++ [35] | 66.53 | 51.58 |
| PointCNN | 70.57 | **59.13** |

Table 3: Sketch classification results.

| Method | MNIST | CIFAR10 |
|---|---|---|
| LeNet [26] | 99.20 | 84.07 |
| Network in Network [29] | 99.53 | **91.20** |
| PointNet++ [33] | 99.49 | 10.0[3] |
| PointCNN | **99.54** | 80.22 |

Table 4: Image classification results.

## 4.2 Ablation Experiments and Visualizations

**Ablation test of the core $\mathcal{X}$-Conv operator.** To verify the effectiveness of the $\mathcal{X}$-transformation, we propose PointCNN without it as a baseline, where lines 4-6 of Algorithm 1 are replaced by $\mathbf{F}_p \leftarrow$ Conv$(\mathbf{K}, \mathbf{F}_*)$. Compared with PointCNN, the baseline has less trainable parameters, and is more "shallow" due to the removal of $MLP(\cdot)$ in line 4 of Algorithm 1. For a fair comparison, we further propose PointCNN w/o $\mathcal{X}$-W/D, which is wider/deeper, and has approximately the same amount of parameters as PointCNN. The model depth of PointCNN w/o $\mathcal{X}$ (deeper) also compensates for

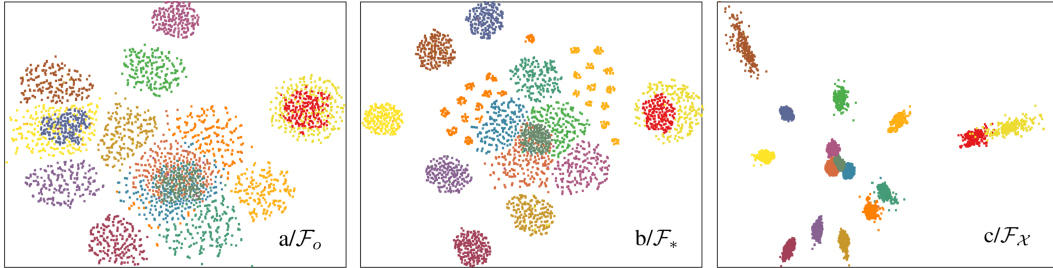

Figure 5: T-SNE visualization of features without (a/$\mathcal{F}_o$), before (b/$\mathcal{F}_*$) and after (c/$\mathcal{F}_\mathcal{X}$) $\mathcal{X}$-transformation.

the decrease in depth caused by the removal of $MLP(\cdot)$ from PointCNN. The comparison results are summarized in Table 5. Clearly, PointCNN outperforms the proposed variants by a significant margin, and the gap between PointCNN and PointCNN w/o $\mathcal{X}$ is not due to model parameter number, or model depth. With these comparisons, we conclude that $\mathcal{X}$-Conv is the key to the performance of PointCNN.

**Visualization of $\mathcal{X}$-Conv features.** Each representative point, with its neighboring points in a particular order, has a corresponding $\mathbf{F}_*$ and $\mathbf{F}_\mathcal{X}$ in $\mathbb{R}^{K\times C}$, where $C = C_\delta + C_1$. For the same representative point, if its neighboring points in different orders are fed into the network, we get a set of $\mathbf{F}_*$ and $\mathbf{F}_\mathcal{X}$, and we denote them as $\mathcal{F}_*$

|  | PointCNN | w/o $\mathcal{X}$ | w/o $\mathcal{X}$-W | w/o $\mathcal{X}$-D |
|---|---|---|---|---|
| Core Layers | $\mathcal{X}$-Conv$\times$4 | Conv$\times$4 | Conv$\times$4 | Conv$\times$5 |
| # Parameter | 0.6M | 0.54M | 0.63M | 0.61M |
| Accuracy (%) | **92.2** | 90.7 | 90.8 | 90.7 |

Table 5: Ablation tests on ModelNet40.

and $\mathcal{F}_\mathcal{X}$. Similarly, we define the set of $\mathbf{F}_*$ in PointCNN w/o $\mathcal{X}$ as $\mathcal{F}_o$. Clearly, $\mathcal{F}_*$ can be quite scattering in the $\mathbb{R}^{K\times C}$ space, since differences in input point order will result in a different $\mathbf{F}_*$. On the other hand, if the learned $\mathcal{X}$ can perfectly canonize $\mathbf{F}_*$, $\mathcal{F}_\mathcal{X}$ is supposed to stay at a canonical point in the space.

To verify this, we show T-SNE visualization of $\mathcal{F}_o$, $\mathcal{F}_*$ and $\mathcal{F}_\mathcal{X}$ of 15 randomly picked representative points from the ModelNet40 dataset in Figure 5, each with one color, and consistent in the sub-figures. Note that $\mathcal{F}_o$ is quite "blended", which indicates that the features from different representative points are not discriminative against each other (Figure 5a). While $\mathcal{F}_*$ is better than $\mathcal{F}_o$, it is still "fuzzy" (Figure 5b). In Figure 5c, $\mathcal{F}_\mathcal{X}$ are "concentrated" by $\mathcal{X}$, and the features of each representative point become highly discriminative. To give an quantitative reference of the "concentration" effect, we firstly compute the feature centers of different representative points, then classify all the feature points to the representative points they belong to, based on nearest search to the centers. The classification accuracies are 76.83%, 89.29% and 94.72% for $\mathcal{F}_o$, $\mathcal{F}_*$ and $\mathcal{F}_\mathcal{X}$, respectively. With the qualitative visualization and quantitative investigation, we conclude that though the "concentration" is far from reaching a point, the improvement is significant, and it explains the performance of PointCNN in feature learning.

| Methods | | PointNet [33] | PointNet++ [35] | 3DmFV-Net [4] | DGCNN [50] | SpecGCN [46] | PCNN [3] | PointCNN |
|---|---|---|---|---|---|---|---|---|
| Parameters | | 3.48M | 1.48M | 45.77M | 1.84M | 2.05M | 8.2M | **0.6M** |
| FLOPs | Training | 43.82B | 67.94B | 48.57B | 131.37B | 49.97B | **6.49B** | 93.03B |
| | Inference | 14.70B | 26.94B | 16.89B | 44.27B | 17.79B | **4.70B** | 25.30B |
| Time | Training | 0.068s | 0.091s | 0.101s | 0.171s | 14.640s | 0.476s | **0.031s** |
| | Inference | 0.015s | 0.027s | 0.039s | 0.064s | 11.254s | 0.226s | **0.012s** |

Table 6: Parameter number, FLOPs and running time comparisons.

**Optimizer, model size, memory usage and timing.** We implemented PointCNN in tensorflow [1], and use ADAM optimizer [21] with an initial learning rate 0.01 for the training of our models. As shown in Table 6, we summarize our running statistics based with the model for classification with batch size 16, 1024 input points on nVidia Tesla P100 GPU, in comparison with several other methods. PointCNN achieves 0.031/0.012 second per batch for training/inference on this setting. In addition, the model for segmentation with 2048 input points has 4.4M parameters runs on nVidia Tesla P100 with batch size 12 at 0.61/0.25 second per batch for training/inference.

# 5 Conclusion

We proposed PointCNN, which is a generalization of CNN into leveraging spatially-local correlation from data represented in point cloud. The core of PointCNN is the $\mathcal{X}$-Conv operator that weights and permutes input points and features before they are processed by a typical convolution. While $\mathcal{X}$-Conv is empirically demonstrated to be effective in practice, a rigorous understanding of it, especially when being composited into a deep neural network, is still an open problem for future work. It is also interesting to study how to combine PointCNN and image CNNs to jointly process paired point clouds and images, probably at the early stages. We open source our code at https://github.com/yangyanli/PointCNN to encourage further development.

### Acknowledgments

Yangyan would like to thank Leonidas Guibas from Stanford University and Mike Haley from Autodesk Research for insightful discussions, and Noa Fish from Tel Aviv University and Thomas Schattschneider from Technical University of Hamburg for proof reading. The work is supported in part by National Key Research and Development Program of China grant No. 2017YFB1002603, the National Basic Research grant (973) No. 2015CB352501, National Science Foundation of China General Program grant No. 61772317, and "Qilu" Young Talent Program of Shandong University.

## Footnotes

*Part of the work was done during Yangyan's Autodesk Research 2017 summer visit.

[2]Actually, this is a special instance of convolution — a convolution that is applied in **one** spatial location. For simplicity, we call it convolution as well.

[3]PointNet++ performs no better than random choice on CIFAR10. We suspect the reason is that, in PointNet++, the RGB features become in-discriminative after being processed by the max-pooling. Together with the lack of "shape" information, PointNet++ fails completely on this task.

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
