[Supplementary Material · PointCNN_supp.pdf]

# PointCNN Supplementary Material

## 1 $\mathcal{X}$-Conv Details

We implement $MLP_\delta(\cdot)$ in Line 2 of Algorithm 1 with two fully connected (FC) layers, each followed by ELU [7] activation function and batch normalization (BN) [19], i.e., $FC(3, C_\delta) \rightarrow ELU \rightarrow BN \rightarrow FC(C_\delta, C_\delta) \rightarrow ELU \rightarrow BN$. We set $C_\delta$ to $C_1/4$.

$MLP(\cdot)$ in Line 4 of Algorithm 1 can be implemented in a similar way: $FC(3*K, K*K) \rightarrow ELU \rightarrow BN \rightarrow FC(K*K, K*K) \rightarrow ELU \rightarrow BN \rightarrow FC(K*K, K*K) \rightarrow BN \rightarrow Reshape(K*K, K \times K)$, where $K*K$ denotes a vector in $\mathbb{R}^{K*K}$, and $K \times K$ denotes a $K$-by-$K$ square matrix. However, this implementation results in $O(K^4)$ parameters. To reduce the parameter number and computation, as well as the overfitting risk due to a large number of parameters, we propose to replace $FC(K*K, K*K)$ with a depthwise convolution $DC(R \times C, C*F)$, which applies $F$ different filters to each of the $C$ column of the input $R \times C$ matrix, yielding a $C*F$ vector, where $R*C*F$ parameters are involved. More specifically, $MLP(\cdot)$ is implemented with $FC(3*K, K*K) \rightarrow ELU \rightarrow BN \rightarrow Reshape(K*K, K \times K) \rightarrow DC(K \times K, K*K) \rightarrow ELU \rightarrow BN \rightarrow Reshape(K*K, K \times K) \rightarrow DC(K \times K, K*K) \rightarrow BN \rightarrow Reshape(K*K, K \times K)$, which results in $O(K^3)$ parameters.

$Conv(\cdot, \cdot)$ in Line 6 of Algorithm 1, if implemented with typical convolution, has $K*(C_1 + C_\delta)*C_2$ trainable parameters. We implemented it with separable convolution [6], which has $K*(C_1 + C_\delta)*DM + (C_1 + C_\delta)*DM*C_2$ trainable parameters, where $DM$ is the depth multiplier, and we use $DM = \lceil C_2/(C_1 + C_\delta) \rceil$ in our implementation. Separable convolution reduces both parameter number and computation compared with that of a typical convolution.

$MLP_g(\cdot)$ is used to harvest the global position information of the representative points in the last $\mathcal{X}$-Conv layer. It is implemented similar to $MLP_\delta(\cdot)$, i.e., $FC(3, C_g) \rightarrow ELU \rightarrow BN \rightarrow FC(C_g, C_g) \rightarrow ELU \rightarrow BN$. We set $C_g$ to $C_2/4$. The $C_g$ dimensional output of $MLP_g(\cdot)$ is concatenated with the $C_2$ dimensional output of the last $\mathcal{X}$-Conv layer for further processing.

In our implementation, a $K$ nearest neighbor search is applied for extracting the $K$ neighboring points. This assumes a more or less uniform distribution of input points. For point clouds with non-uniform distribution, a radius search can be applied first, and then $K$ points can be randomly picked out of the radius search results.

In theory, the $\mathcal{X}$-transformation can be applied on either the features or the kernels. We opt to apply it on the features, in which way the follow up operation is a standard convolution operation that is highly optimized by popular deep learning frameworks.

## 2 Dataset Details

We conducted extensive evaluation of PointCNN on datasets of various types and scales. Here we introduce the details of the datasets, as well as how we pre-process and feed them into PointCNN:

- Object datasets: ModelNet40 [52] and ShapeNet Parts [54].
    - ModelNet40 is composed of 12311 3D mesh models from 40 categories, with a 9843/2468 training/testing split. Both the gravity and "facing" directions of the models are mostly aligned in the dataset. In the "Pre-aligned" setting, the models are used for training and testing, without random horizontal rotations. In which way, the relative consistent "facing" direction is leveraged by the network. In the "Unaligned" setting, random horizontal rotations are explicitly applied on either the training or the testing models, **not** as a data augmentation, but to "forget" the relative consistent "facing" directions thus better approximate the scenarios in real world applications, where the "facing" direction of the objects are often unknown. We use the point cloud conversion of ModelNet40 provided by [33] as our input, where 2048 points are sampled from each mesh, and we further sample $\mathcal{N}(1024, 128^2)$ points to train a model for testing with 1024 points on the classification task.
    - ShapeNet Parts contains 16880 models (14006/2874 training/testing split) from 16 shape categories, each annotated with 2 to 6 parts and there are 50 different parts in total. Each point sampled from the models is associated with a part label. The task is to predict the part label for each point, thus a segmentation task, and can be treated as a dense point-wise classification problem. The category label for each model is given, and can be used for trimming irrelevant predictions, same as that in [12]. $\mathcal{N}(2048, 256^2)$ points are sampled from each point cloud to train a model for testing with 2048 input points on the segmentation task. Each testing point cloud is sampled multiple times to make sure all the points are evaluated at least $r$ ($r = 10$ in our experiments) times at testing time.
- Indoor scene datasets: S3DIS [2] and ScanNet [9]. While ModelNet40 and ShapeNet models are mostly made by 3D modeling tools, S3DIS and ScanNet are from real scans of indoor environments.

Figure 1: PointCNN model zoo, where (a) is used for ModelNet40 (channel number in bold) and ScanNet classification, (b) is used for TU-Berlin sketch classification, (c) is used for Quick Draw sketch classification, (d) is used for ScanNet and S3DIS segmentation, and (e) is used for ShapeNet Parts segmentation.

- S3DIS contains 3D scans from Matterport scanners in 6 areas including 271 rooms. Each point with RGB features in the scan is annotated with one of the semantic labels from 13 categories. The task is segmentation. The data is firstly split by room, and then the rooms are sliced into 1.5m by 1.5m blocks, with 0.3m padding on each side. The points in the padding areas serve as context of the internal points, and themselves are not linked to loss in the training phase, nor used for prediction in the testing phase. Each block is moved to a local coordinate system defined by its center. Random horizontal rotations are applied on the sliced blocks for data augmentation. The rotated blocks are handled in the same way as the object point clouds in ShapeNet Parts.

- ScanNet contains 1513 scanned and reconstructed indoor scenes, with $1201/312$ scenes for training/testing in semantic voxel labeling of 17 categories. We firstly prepare data in the same way as that of S3DIS to train a segmentation model, and the segmentation results on testing data are then converted into semantic voxel labeling, as that in [35], for a fair comparison with previous methods. The $9305/2606$ training/testing object instances from the 17 categories in ScanNet are also used for evaluating classification task. Note that ScanNet comes with RGB information for each point. However, they are not used in previous methods. To make fair comparisons, we do not use them either.

- 2D sketch datasets: TU-Berlin [11] and Quick Draw [15]. Similar to surfaces in 3D space, line sketches in 2D are inherently of less dimension than the ambient space, and can be represented as point cloud, thus we consider 2D sketches good arena for evaluating neural networks that are designed to consume point cloud data. TU-Berlin has sketches from 250 categories, with 80 sketches from each category, where $2/3$ are used for training and the rest $1/3$ for testing. Quick Draw is the largest available sketch dataset, with sketches from 345 categories, each with $70000/2500$ training/testing samples. We sample $\mathcal{N}(512, 64^2)$ points from the sketch stokes to train a model for testing with 512 points on sketch classification task.

- Image datasets: MNIST and CIFAR10. MNIST and CIFAR10 are widely used for sanity check of image CNNs. Since PointCNN is a generalization of CNNs, we would like to evaluate PointCNN on the point cloud representation of MNIST and CIFAR10. For MNIST, we randomly sample 160 **foreground** pixels and convert them into point cloud representation, with the gray-scale pixel value as the input feature. For CIFAR10, we randomly sampled 512 pixels out of the $32 \times 32$ pixels for converting into point cloud with RGB features. Note that there is "shape" information in the MNIST point cloud, sine the point cloud follow the digits' structure, but this is not the case for the CIFAR10 point cloud, where the points are mostly the same blob for all the data samples.

# 3 PointCNN Model Zoo

In Figure 1, we list the PointCNNs used for classification and segmentation tasks on multiple benchmark datasets. PointCNNs are easy to implement, setup, and tune. Larger $C$ are used for layers with more abstract/semantic information, such as the top layers in classification networks, and middle layers in "Conv-DeConv" segmentation networks. To relax the memory demand, smaller $K$s are used at layers with large number of representative points, such as bottom layers of classification networks, and top and bottom layers of segmentation networks. Deeper PointCNN with larger receptive field in the last $\mathcal{X}$-Conv layer are used for larger or harder datasets. The skip-links, together with the dilation parameter $D$, make it easy to fuse information from different scales (receptive fields), as illustrated in (d) and (e), which is essential for segmentation tasks.

| Method | pIoU | mpIoU | air plane | bag | cap | car | chair | ear phone | guitar | knife | lamp | laptop | motor bike | mug | pistol | rocket | skate board | table |
|---|---|---|---|---|---|---|---|---|---|---|---|---|---|---|---|---|---|---|
| SyncSpecCNN [55] | 84.74 | 82.0 | 81.55 | 81.74 | 81.94 | 75.16 | 90.24 | 74.88 | **92.97** | 86.1 | 84.65 | 95.61 | 66.66 | 92.73 | 81.61 | 60.61 | 82.86 | 82.13 |
| Pd-Network [22] | 85.49 | 82.7 | 83.31 | 82.42 | 87.04 | 77.92 | 90.85 | 76.31 | 91.29 | 87.25 | 84.0 | 95.44 | 68.71 | 94.0 | 82.9 | 62.97 | 76.44 | 83.18 |
| SSCN [12] | 85.98 | 83.3 | 84.09 | 82.99 | 83.97 | 80.82 | **91.41** | 78.16 | 91.6 | **89.1** | 85.04 | 95.78 | 73.71 | 95.23 | 84.02 | 58.53 | 76.02 | 82.65 |
| SpiderCNN [53] | 85.3 | 81.7 | 83.5 | 81 | 87.2 | 77.5 | 90.7 | 76.8 | 91.1 | 87.3 | 83.3 | 95.8 | 70.2 | 93.5 | 82.7 | 59.7 | 75.8 | 82.8 |
| SO-Net [27] | 84.9 | 81.0 | 82.8 | 77.8 | **88.0** | 77.3 | 90.6 | 73.5 | 90.7 | 83.9 | 82.8 | 94.8 | 69.1 | 94.2 | 80.9 | 53.1 | 72.9 | 83.0 |
| PCNN [3] | 85.1 | 81.8 | 82.4 | 80.1 | 85.5 | 79.5 | 90.8 | 73.2 | 91.3 | 86.0 | 85.0 | 95.7 | 73.2 | 94.8 | 83.3 | 51.0 | 75.0 | 81.8 |
| KCNet [42] | 83.7 | 82.2 | 82.8 | 81.5 | 86.4 | 77.6 | 90.3 | 76.8 | 91.0 | 87.2 | 84.5 | 95.5 | 69.2 | 94.4 | 81.6 | 60.1 | 75.2 | 81.3 |
| Kd-Net [22] | 82.3 | 77.4 | 80.1 | 74.6 | 74.3 | 70.3 | 88.6 | 73.5 | 90.2 | 87.2 | 81.0 | 94.9 | 57.4 | 86.7 | 78.1 | 51.8 | 69.9 | 80.3 |
| 3DmFV-Net [4] | 84.3 | 81.0 | 82.0 | 84.3 | 86.0 | 76.9 | 89.9 | 73.9 | 90.8 | 85.7 | 82.6 | 95.2 | 66.0 | 94.0 | 82.6 | 51.5 | 73.5 | 81.8 |
| RSNet [18] | 84.9 | 81.4 | 82.7 | 86.4 | 84.1 | 78.2 | 90.4 | 69.3 | 91.4 | 87.0 | 83.5 | 95.4 | 66.0 | 92.6 | 81.8 | 56.1 | 75.8 | 82.2 |
| DGCNN [50] | 85.1 | 82.3 | **84.2** | 83.7 | 84.4 | 77.1 | 90.9 | 78.5 | 91.5 | 87.3 | 82.9 | 96.0 | 67.0 | 93.3 | 82.6 | 59.7 | 75.5 | 82.0 |
| PointNet [33] | 83.7 | 80.4 | 83.4 | 78.7 | 82.5 | 74.9 | 89.6 | 73.0 | 91.5 | 85.9 | 80.8 | 95.3 | 65.2 | 93.0 | 81.2 | 57.9 | 72.8 | 80.6 |
| PointNet++ [35] | 85.1 | 81.9 | 82.4 | 79.0 | 87.7 | 77.3 | 90.8 | 71.8 | 91.0 | 85.9 | 83.7 | 95.3 | 71.6 | 94.1 | 81.3 | 58.7 | 76.4 | 82.6 |
| SGPN [49] | 85.8 | 82.8 | 80.4 | 78.6 | 78.8 | 71.5 | 88.6 | 78 | 90.9 | 83 | 78.8 | 95.8 | **77.8** | 93.8 | **87.4** | 60.1 | **92.3** | **89.4** |
| PointCNN | **86.14** | **84.6** | 84.11 | **86.47** | 86.04 | **80.83** | 90.62 | **79.70** | 92.32 | 88.44 | 85.31 | **96.11** | 77.20 | **95.28** | 84.21 | **64.23** | 80.00 | 82.99 |

Table 1: Segmentation result comparisons on ShapeNet Parts [54] in part-averaged IoU (pIoU, %) , mean per-class pIoU (mpIoU, %) and per-class pIoU (%).

| Method | OA | mAcc | mIoU | ceiling | floor | wall | beam | column | window | door | table | chair | sofa | bookcase | board | clutter |
|---|---|---|---|---|---|---|---|---|---|---|---|---|---|---|---|---|
| PointNet [33] | 78.5 | 66.2 | 47.6 | 88.0 | 88.7 | 69.3 | 42.4 | 23.1 | 47.5 | 51.6 | 54.1 | 42.0 | 9.6 | 38.2 | 29.4 | 35.2 |
| SPGraph [24] | 85.5 | 73.0 | 62.1 | 89.9 | 95.1 | 76.4 | 62.8 | 47.1 | 55.3 | **68.4** | **73.5** | **69.2** | **63.2** | 45.9 | 8.7 | 52.9 |
| RSNet [18] | - | 66.45 | 56.47 | 92.48 | 92.83 | **78.56** | 32.75 | 34.37 | 51.62 | 68.11 | 60.13 | 59.72 | 50.22 | 16.42 | 44.85 | 52.03 |
| PointCNN | **88.14** | **75.61** | **65.39** | **94.78** | **97.3** | 75.82 | **63.25** | **51.71** | **58.38** | 57.18 | 71.63 | 69.12 | 39.08 | **61.15** | **52.19** | **58.59** |

Table 2: Segmentation result comparisons on the S3DIS [2] dataset in overall accuracy (OA, %), micro-averaged accuracy (mAcc, %), micro-averaged IoU (mIoU, %) and per-class IoU (%).

| Method | OA | mAcc | mIoU | ceiling | floor | wall | beam | column | window | door | table | chair | sofa | bookcase | board | clutter |
|---|---|---|---|---|---|---|---|---|---|---|---|---|---|---|---|---|
| PointNet [33] | - | 48.98 | 41.09 | 88.80 | 97.33 | 69.80 | 0.05 | 3.92 | 46.26 | 10.76 | 58.93 | 52.61 | 5.85 | 40.28 | 26.38 | 33.22 |
| SPGraph [24] | **86.38** | 66.50 | 58.04 | 89.35 | 96.87 | 78.12 | 0.00 | **42.81** | 48.93 | 61.58 | **84.66** | 75.41 | **69.84** | 52.60 | 2.10 | 52.22 |
| SegCloud [45] | - | 57.35 | 48.92 | 90.06 | 96.05 | 69.86 | 0.00 | 18.37 | 38.35 | 23.12 | 70.40 | 75.89 | 40.88 | 58.42 | 12.96 | 41.60 |
| PCCN [48] | - | **67.01** | **58.27** | 92.26 | 96.20 | 75.89 | **0.27** | 5.98 | **69.49** | **63.45** | 66.87 | 65.63 | 47.28 | **68.91** | 59.10 | 46.22 |
| PointCNN | 85.91 | 63.86 | 57.26 | **92.31** | **98.24** | **79.41** | 0.00 | 17.60 | 22.77 | 62.09 | 74.39 | **80.59** | 31.67 | 66.67 | **62.05** | **56.74** |

Table 3: Segmentation result comparisons on the S3DIS [2] Area 5 in overall accuracy (OA, %), micro-averaged accuracy (mAcc, %), micro-averaged IoU (mIoU, %) and per-class IoU (%).

# 4 Detailed Segmentation Results

We show detailed segmentation result comparisons on ShapeNet Parts in Table 1, we can see our approach achieves the best overall performance and are best on 7 of the 16 categories.

We show detailed segmentation result comparisons on S3DIS in Table 2, we can see our approach achieves the best overall performance and are best on 6 of the 13 categories. The detailed segmentation result comparisons on S3DIS Area 5 are summarized in Table 3, as some of the literatures only report the performance on this area.