[Reviews · NeurIPS 2018]

Reviewer 1



This paper addresses the problem of representing unordered point sets for recognition applications. The key insights is a “chi-convolution” operator that learns to “permute" local points and point-features into a canonical order within a neural network. The approach is demonstrated on 3D point cloud recognition and segmentation and 2D sketch and image classification applications. Positives: The paper addresses a known hard problem - how to properly represent unordered point sets for recognition. As far as I’m aware, the paper describes a novel and interesting approach for learning to “permute" local point neighborhoods in unordered point sets. The paper demonstrates strong results for a variety of tasks over different datasets and appears to perform as well or better than very recent approaches. As far as I can see, the paper cites well prior work in this area. The approach is reasonably well described. Negatives: I have a few concerns: A. The title and introduction are not clear. The title is not descriptive and is too similar to “PointNet”. To me, the main focus is on learning to permute the ordering of local point sets, so perhaps the title could be updated to reflect this. The introduction does not convey well the main contribution of the work. Also, Fig 1 showing the toy example was very confusing to me on first reading. For instance, what do the lines next to the points mean? Also, Equations (1a) and (1b) were not helpful to me. I think clearly illustrating and describing in words the permutation issue would be sufficient. Moreover, explicitly stating the contribution would be great. B. The proposed hierarchical convolution technique (Section 3.1) is incremental to prior work (e.g., PointNet++). My suggestion is to move this later to “implementation details” and focus on the chi-convolution in Section 3. C. The paper gives an intuition that the chi-convolution operator is learning permutation, but this is not explicitly demonstrated. There is some qualitative notion in the tsne plots in Fig 5, but it would be great to see this quantitatively somehow. One possibility is to show how sensitive the results are to local permutation changes in the test set (e.g., with error bars). Overall, I think the results are very promising and the technique would be of interest to the NIPS community. It looks like source code will be released, so results should be reproducible. I’m inclined to recommend "accept as poster". Minor comments: + L121-123: This wasn’t clear to me how it’s implemented. Since source code will be released, this is not a major concern. One possibility is to add more details in the supplemental. + Algorithm 1 L1: The local points are centered relative to p. Is there any sensitivity to scaling? + L133: f_i is not redefined as p_i is. + Fig 5: I was confused at first as to what the three plots corresponded to. Perhaps include F_o, F_*, F_x below each plot.

Reviewer 2



This paper proposed a generalized CNN to point-cloud which preserves the geometry shape information, compared to the existing methods. It introduces an X-transformation that aggregating the features, which can leverage both the feature the coordinate information. Strength: This paper presents a clear motivation and solution, with comprehensive experimental validations and ablation studies. Weakness: I have several technical concerns. First, the presentation of the insight of (2) is not clear enough to me. It is described as "weighted" and "permuted", very briefly in 3.2. However, the formulation reminds me of several recent attention works [1][2], or more traditionally, the thin plate spline (TPS) with the 1st order case. The transformation X is more like a K*K affinity matrix where each entry represents some certain measurement (e.g., geometry closeness, or please check the radial basis function in TPS, where your paper provides a learnable such function by 4 in Algo.1) between a pair of points. Since X is a coordinate affinity and F_{*} contains the coordinate information, the step-5 in Algo. 1 actually represents a sense of linear spatial transformation. It is a nice formulation that corresponds to the concept of "transfer points to canonical space", however, the author did not go deep into it. I hope the author can provide more analysis of (2), or correct me if the above is wrong. Second, I am not sure if the authors made some contradictory statements between line 156 (dependent on the order), and line 46 (invariant to the ordering)? There is another isolated statement (line 134) that S is an unordered set, why? My understanding towards this is that operations are dependent in any case to the ordering even if the points are in a canonical space, or in an image. Third, the improvements over other sota methods are not obvious, and the results w.r.t PointNet++ in Cifar10 still does not make much sense to me. Given the above strength and weakness, I would rate this work as 6: Marginally above the acceptance threshold. [1] Vaswani et al., Attention is all you need. NIPS 2017. [2] Wang et al., Non-local Neural Networks. CVPR 2018. Update: I would not change the score after reading the rebuttal. However, I still insist that this paper should be improved in terms of theoretical analysis, as a NIPS paper, instead of explaining that some future work will be delivered.

Reviewer 3



Paper summary: - This work proposes a new neural network layer to perform convolution-like operations on unordered point cloud data. The main crux of the technique is the transformation of given unordered points into a canonical representation using a learned matrix multiplication and then a standard convolution is applied on those canonically represented points. Wide range of experiments on classification and segmentation tasks validate the use of the proposed technique in comparison to existing techniques. Strengths: - Simple yet generic technique for processing point cloud data with unordered points. - Applications on both classification and segmentation on a wide range of datasets. - Very good performance compared to existing state-of-the-art techniques. Weaknesses: - A main weakness of this work is its technical novelty with respect to spatial transformer networks (STN) and also the missing comparison to the same. The proposed X-transformation seems quite similar to STN, but applied locally in a neighborhood. There are also existing works that propose to apply STN in a local pixel neighborhood. Also, PointNet uses a variant of STN in their network architecture. In this regard, the technical novelty seems limited in this work. Also, there are no empirical or conceptual comparisons to STN in this work, which is important. - There are no ablation studies on network architectures and also no ablation experiments on how the representative points are selected. - The runtime of the proposed network seems slow compared to several recent techniques. Even for just 1K-2K points, the network seem to be taking 0.2-0.3 seconds. How does the runtime scales with more points (say 100K to 1M points)? It would be good if authors also report relative runtime comparisons with existing techniques. Minor corrections: - Line 88: "lose" -> "loss". - line 135: "where K" -> "and K". Minor suggestions: - "PointCNN" is a very short non-informative title. It would be good to have a more informative title that represents the proposed technique. - In several places: "firstly" -> "first". - "D" is used to represent both dimensionality of points and dilation factor. Better to use different notation to avoid confusion. Review summary: - The proposed technique is sensible and the performance on different benchmarks is impressive. Missing comparisons to established STN technique (with both local and global transformations) makes this short of being a very good paper. After rebuttal and reviewer discussion: - I have the following minor concerns and reviewers only partially addressed them. 1. Explicit comparison with STN: Authors didn't explicitly compare their technique with STN. They compared with PointNet which uses STN. 2. No ablation studies on network architecture. 3. Runtimes are only reported for small point clouds (1024 points) but with bigger batch sizes. How does runtime scale with bigger point clouds? Authors did not provide new experiments to address the above concerns. They promised that a more comprehensive runtime comparison will be provided in the revision. Overall, the author response is not that satisfactory, but the positive aspects of this work make me recommend acceptance assuming that authors would update the paper with the changes promised in the rebuttal. Authors also agreed to change the tile to better reflect this work.